# SCREENER: Streamlined collaborative learning of NER and RE model for discovering gene-disease relations

**Minjun Park**[1☯], **Chan Ung Jeong**[1☯], **Young Sang Baik**[2☯], **Dong Geon Lee**[1☯], **Jeong U. Park**[2], **Hee Jung Koo**[1], **Tae Yong Kim**[1]*

**1** Standigm Inc., Seoul, South Korea, **2** SK Inc., C&C, Seongnam, South Korea

☯ These authors contributed equally to this work.
* taeyong.kim@standigm.com

**Data Availability Statement:** All data files are available from the Zenodo database (accession number 8385406).

## Abstract

Finding relations between genes and diseases is essential in developing a clinical diagnosis, treatment, and drug design for diseases. One successful approach for mining the literature is the document-based relation extraction method. Despite recent advances in document-level extraction of entity-entity, there remains a difficulty in understanding the relations between distant words in a document. To overcome the above limitations, we propose an AI-based text-mining model that learns the document-level relations between genes and diseases using an attention mechanism. Furthermore, we show that including a direct edge (DE) and indirect edges between genetic targets and diseases when training improves the model's performance. Such relation edges can be visualized as graphs, enhancing the interpretability of the model. For the performance, we achieved an F1-score of 0.875, outperforming state-of-the-art document-level extraction models. In summary, the SCREENER identifies biological connections between target genes and diseases with superior performance by leveraging direct and indirect target-disease relations. Furthermore, we developed a web service platform named SCREENER (**S**treamlined **C**ollabo**R**ativ**E** l**E**arning of **NE**r and **R**e), which extracts the gene-disease relations from the biomedical literature in real-time. We believe this interactive platform will be useful for users to uncover unknown gene-disease relations in the world of fast-paced literature publications, with sufficient interpretation supported by graph visualizations. The interactive website is available at: https://ican.standigm.com.

## Introduction

Finding the relation between genes and diseases [1–5] is essential in developing clinical diagnoses, treatments, and drug designs for diseases. These gene-disease relations are generally collected from various biomedical literature and built into databases. Due to the vast number of biomedical research publications, identifying gene-disease relations is highly challenging. For example, PubMed Central (PMC), a free professional archive of the biomedical literature, includes over 8 million articles in 2022 [6].

**Funding:** The author(s) received no specific funding for this work.

Since researchers cannot analyze all these reports directly, an interest in automatic analysis technology using natural language processing (NLP) grew rapidly. In biomedical NLP, named entity recognition (NER) and relation extraction (RE) are two challenging tasks in the problem of predicting the relation between genes and diseases. Specifically, the NER module predicts which entity type each tokenized words belong to. After correctly identifying entities of tokenized words, the RE module searches for entities predicted as either genes or diseases, which are grouped into gene-disease pair to determine whether the paired entities have link with one another. In general, extracting relations within a document (document-based) is more challenging than extracting relations between entities within a sentence (sentence-based) as the model loses predictability power as input sequences get longer. Consequently, previous approaches widely used sentence-level methods to extract gene-disease relations [7, 8]. For example, Lee et al. developed the BioBERT-GAD model that learns the relations contained within a single sentence by learning their characteristics [7]. This sentence-level approach inherently has limitations, as gene-disease relations often exist beyond a single sentence. Recently, there have been efforts to develop document-based relation extraction methods to overcome the limitations of sentence-based models [9–16]. For example, Su et al. developed a document-level extraction model called RENET2 [17], improving the previous model RENET [18]. RENET2 trains the model using convolution and recurrent neural networks (CNN and RNN, respectively), which takes tokenized sentences as input. This approach is effective in recognizing words that occur in a domain-specific manner, free from the size of the document as defined by the number of sentences. However, the performance of CNN-RNN training is relatively lower than pre-trained language models due to the commonality of repeated occurrences of words in biomedical literature. Also, the model loses the power to understand the relations between distant words in a document as CNN focuses on local features. Furthermore, RNN suffers from the problem of vanishing gradients when learning long data sequences [19].

Here, we present an end-to-end document-based gene-disease relation extraction model called SCREENER, which collaboratively trains NER and RE modules (See Fig 1). The entire process of SCREENER splits into two parts: 1) the model identifies biological entities such as genes and diseases in a given document (NER) and 2) builds relations classifier from semantic relationship category within a set of artifacts (RE). In training, the loss calculated from the NER module adds to the RE loss, enabling collaborative learning of NER and RE (See Fig 2). While previous models solely relied on indirect edges when training the relation extraction model, we utilize both direct and indirect edges between genes and diseases in training, which elucidates direct relationships between entities and improves the model's performance overall. For usability, we built an interactive web platform that returns gene-disease relations in visual graphs upon users' requests. The web service is available at: https://ican.standigm.com.

## Materials and methods

### Dataset

**AGAC dataset.** For the dataset to train the model, we combined publicly available AGAC and manually annotated the SCREENER dataset. AGAC dataset provides 250 documents with 3,317 entities and 2,729 relations [20] in the training set. In detail, the AGAC dataset provides "trigger words," which refer to the intermediate node that completes the entity-entity relationship. Here, the AGAC relation annotation consists of two predicates, "Theme Of" and "Cause of." The detailed data composition is described in S1 Table. To integrate the AGAC dataset with the SCREENER corpus, we changed relation types "ThemeOf" and "CauseOf" in AGAC to "LinkedOf."

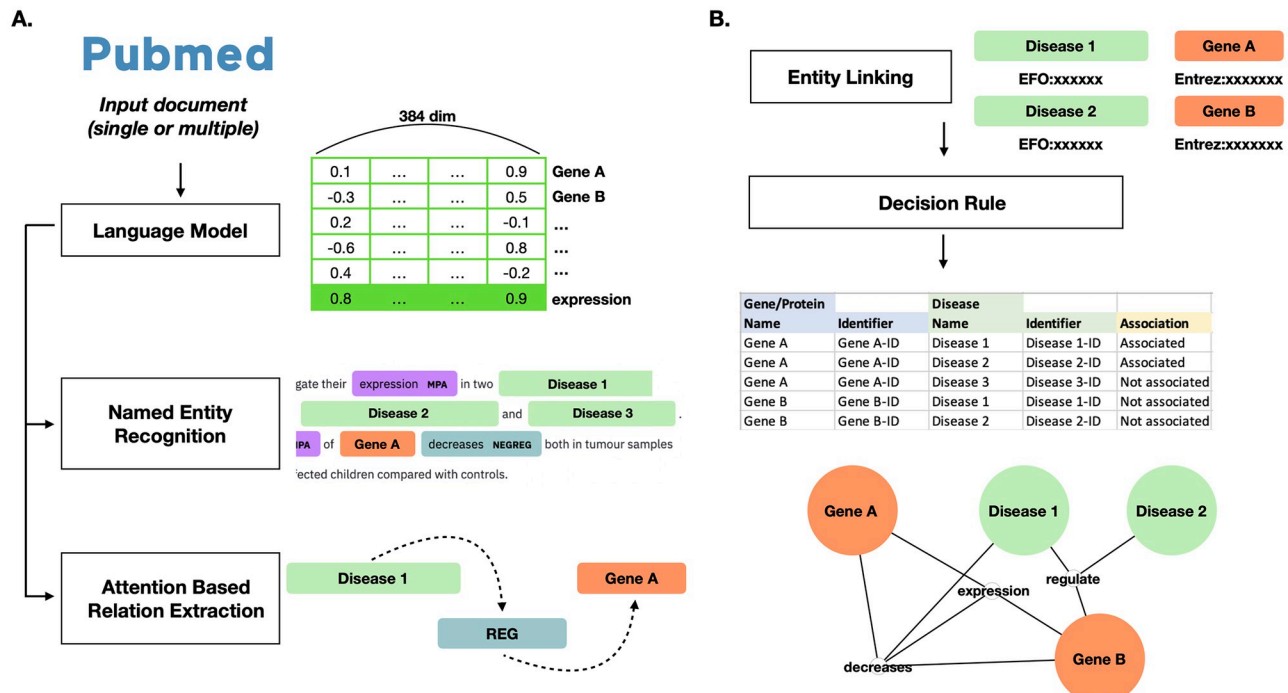

**Fig 1. SCREENER web service workflow.** A simple schematic diagram of SCREENER's architecture is shown. Modules A to E correspond to the actions taken for each input document, and module F integrates results across documents to provide collective relations when the SCREENER has received multiple documents as input.

**SCREENER dataset.** One primary obstacle to extracting gene-disease relation information is the lack of data; Thus, we manually labeled gene-disease interactions in biomedical literature abstracts. This manual work was provided by EXCELRA, a leading informatics service company that offers a workforce of domain expertise in biomedical science.

We define twelve types into two categories: biological entities (Gene, Disease, Protein, Variation, Enzyme, Molecular Physiological Activity, Cellular Physiological Activity, Interaction, Pathway) and regulatory expression (Regulation, Positive Regulation, Negative Regulation). Following the same schema as the AGAC dataset, we label these entities and expressions as trigger words with a single predicate "LinkedOf" (See Fig 3).

In the SCREENER dataset, we introduce a "direct edge (DE)" that directly connects genes and disease (See Fig 4), enabling the model to learn gene-disease relations using both indirect and direct edges. Overall, we annotated 1,377 PubMed articles, including 52,709 entities and 43,601 relations. Among entities, genes and diseases consist of nearly 40 percent of the data (See S1 Fig). Detailed information on the annotation process is described in S3 Text. For open source contribution, the training and evaluation data, and annotation protocol are available at https://doi.org/10.5281/zenodo.8385406.

## SCREENER architecture

The SCREENER consists of two modules: 1) the NER module that utilizes a pre-trained BERT model and 2) the RE module that takes advantage of the NER module components, reinforced with an attention mechanism. In this section, we lay out the components of end-to-end SCREENER architecture.

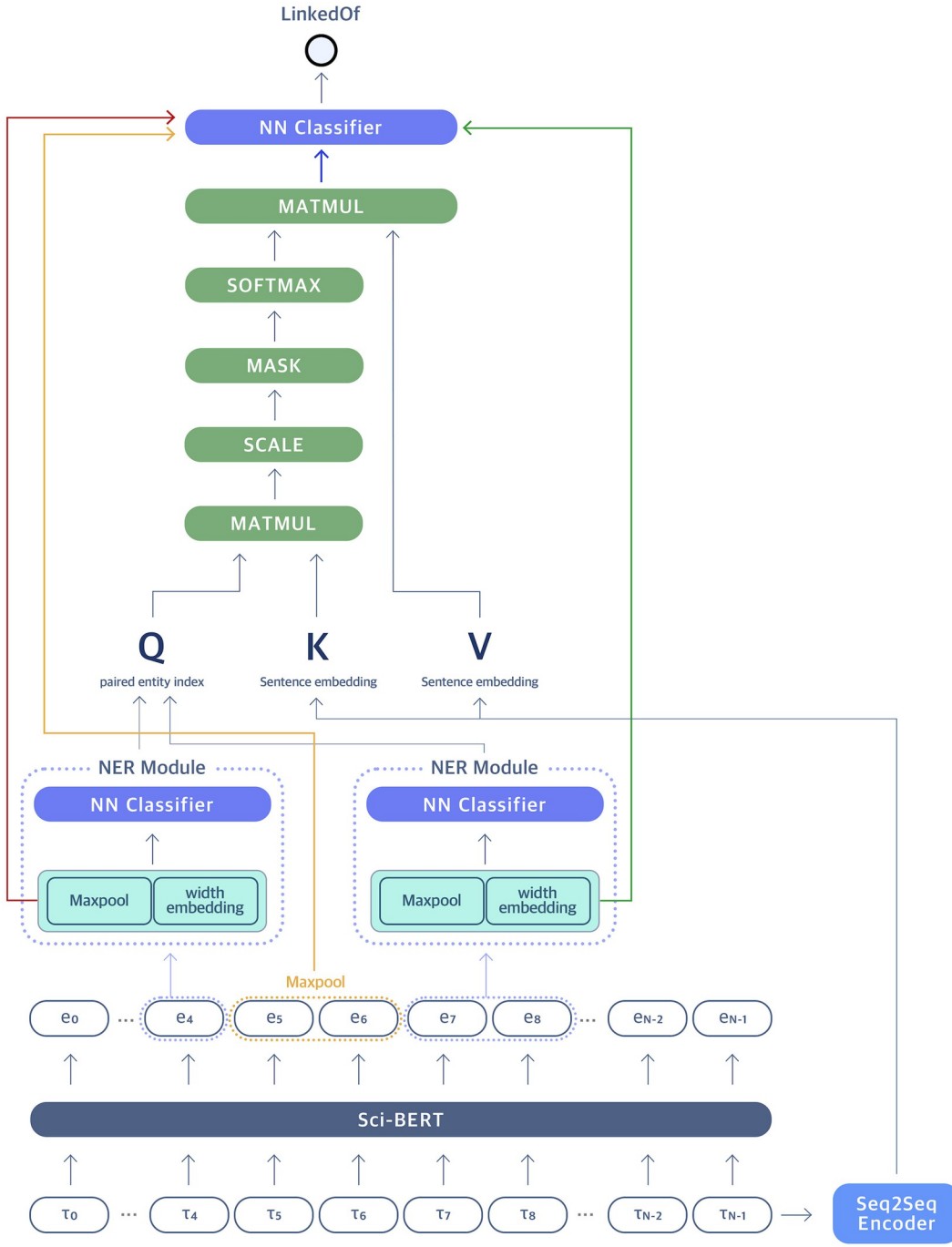

**Fig 2. SCREENER architecture diagram.** The SCREENER consists of two modules: Named Entity Recognition (NER) and Relation Extraction (RE). The NER module utilizes a pre-trained BERT model on scientific texts to predict entity types of the spans. The RE module concatenates four vectors: each span-based entity vector (red and green), a max-pooled vector from embedded tokens positioned between two entities (yellow), and the attention score vector (blue). Together, the model is designed to learn the context features of the input sentences, enhancing the model's performance in predicting the link between two queried entities from the NER module.

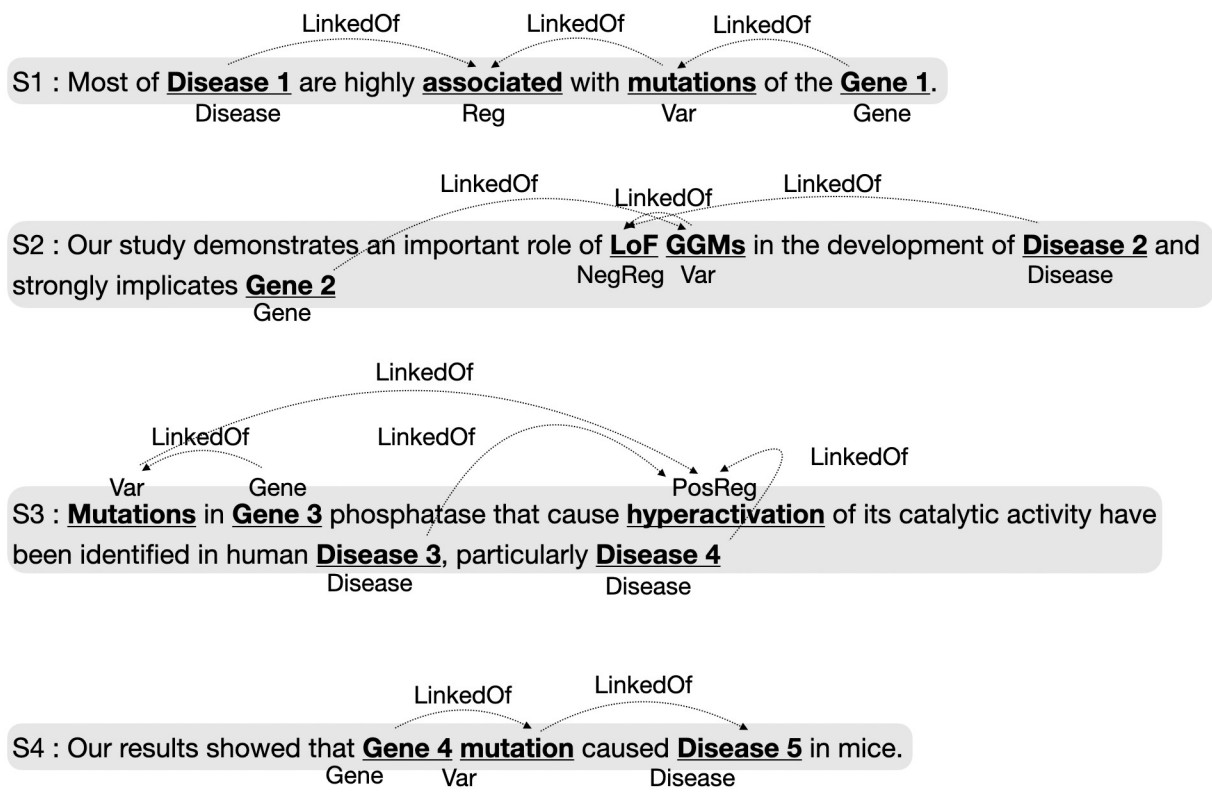

**Fig 3. Example of related entities.** The predefined relation (Linked Of) is represented with above arrows.

**Tokenization and pre-trained model.** For the first step, the collected sentences in the abstracts of scientific texts are tokenized to fit into the model. In detail, the input texts in a biomedical document are tokenized using byte pair encoding (BPE) tokens (i.e., word pieces) [21], which are used to limit vocabulary size and map out-of-vocabulary (OOV) words. The BPE tokens are then passed to the pre-trained language model and transformed into the corresponding hidden state sequence vectors known as knowledge vectors.

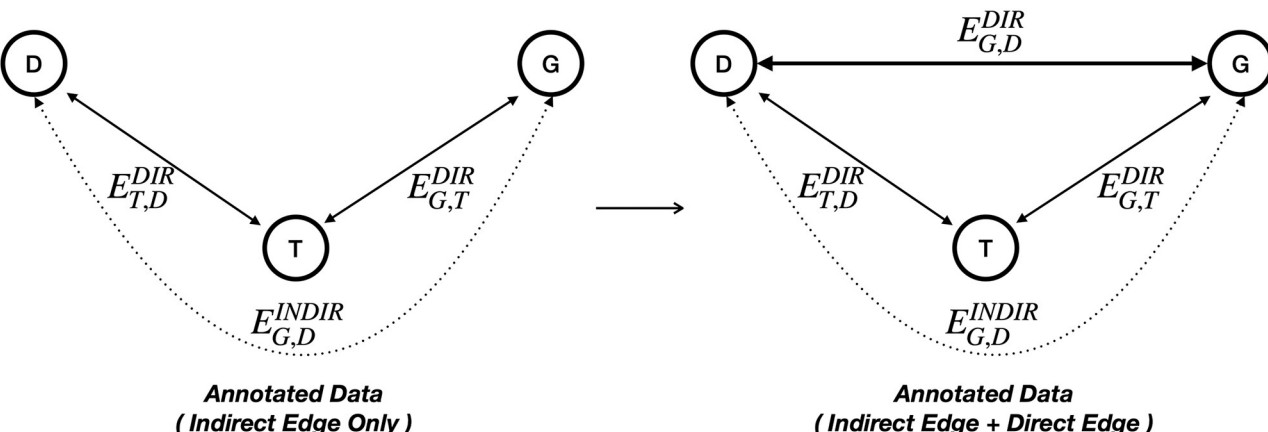

**Fig 4. Indirect edges and direct edge.** Before we add the direct edge between gene and disease, there is only an indirect edges related a given gene with a disease. However, adding direct edge improves performance.

**Named entity recognition module.** In the NER module, the SCREENER adopts the best performing pre-trained BERT model called SciBERT, trained on 1.14 million biomedical literature with 3.1 billion tokens (See Table 2) [22]. We apply span-based sub-sequencing method presented by Eberts et al. [23]. The previous research shows considering all spans in a document as potential entities is highly useful to learn the distributions of the context [24]. For the process, every sub-sequence of the embedded vectors from SciBERT are considered as tokens. Then, max-pooling is applied on each sub-sequence and concatenated with width-embedding vector, which accounts for different width of the tokens. In general, the longer the sub-sequence, the less likely the tokens will be an entity. This concatenated vector is passed to sub-module of ELECTRA [25], which is a shallow neural network module, for fine-tuning using scientific texts from PubMed database (See Fig 2). This shallow classifier recognizes entities in given input sentences and classifies the type of recognized entities. As for training, the model optimizes by minimizing the loss function below:

$$Loss = -\sum_{t=1}^{M} y_{o,t} \log (p_{o,t})$$

where, the classifier learns by calculating probability $p$ to classify $M$ entity types for observation $o$. The $y$ is a binary indicator (0 or 1) if the type label $t$ is the correct classification for observation $o$. In summary, the NER module predicts entity candidates and passes the output to the RE module.

**Relation extraction module.** Once the name entities of the words are defined, the RE module determines if there are valid relationships between distant words across sentences. The notation of the relationship between two entities can be written as a triplet: $(e_0, e_1, r)$. Here, $e_0$, $e_1$ are two entities with relationship type $r$. Once entity types are predicted from the NER module, the RE module concatenates four vectors: each span-based entity vector (red and green), a max-pooled vector from embedded tokens positioned between two entities (yellow), and the attention score vector (blue) (See Fig 2). Here, each of two entity vectors represents concatenated vectors of max-pooling and width embedding from the NER module. Additionally, to consider tokens between two entities of interest (Gene and Disease), we included max-pooled vector of in-between tokens. In Fig 2, max-pooled vector of embedded tokens $(e_5, e_6)$ corresponds to the following vector. For the loss, the model uses binary cross entropy loss from link prediction between paired entities. The final loss is calculated by adding the loss value from the NER module.

**Attention mechanism.** The attention module enables the model to focus on specific parts of the input sentences to enhance model prediction and inference. In other words, an attention mechanism helps a network focus on the most relevant parts of its input rather than processing the entire input equally. Mathematically, an attention mechanism is a function that maps an input and a set of weights to a weighted sum of the input. The attention weights, also known as attention coefficients, represent the relative importance of each input and are typically learned by the network through training. The attention mechanism function can be represented as follows:

$$Attention(X, W) = \sum_{i=1}^{n} w_i * x_i$$

where $X$, $W$, and $n$ represent input to the model, set of attention weights, and the number of input sentence tokens, respectively. In this equation, the attention mechanism first calculates the weighted sum of the tokens by multiplying each element of the input with its corresponding attention weight and summing the results. The network then uses this weighted sum as

input for the next module or to make a final prediction or inference. In the context of sentence learning, tokenized input sentences are encoded using Seq2Seq encoder [26] for the key and value parameters. As for the query parameter, the model takes entity pairs predicted from the NER module. Overall, an attention mechanism allows a neural network to emphasize the essential parts of its input.

## SCREENER entity linking and visualization web service

The purpose of the SCREENER is to facilitate relation extractions in biomedical literature. Thus, we incorporate post-processing modules to meet the specific needs of the users. We offer graph visualizations to allow users to easily follow the web platform at https://ican. standigm.com.

**A. Entity linking module.** The entity linking (EL) module links suitable public identifiers for genes, proteins, and diseases recognized by the NER module. Here, we consider the genes and proteins as possible disease targets and give them corresponding identifiers based on the NCBI Gene [27]. In the case of proteins, the identifiers of the genes encoding the proteins are endowed. Disease identifiers refer to Experimental Factor Ontology [28]. The pipeline component of SpaCy [29] was modified and applied as a knowledge-based entity linking method. The EL module performs a string-overlap-based search (char-3grams) on the named entities and compares them with concepts from dictionaries (i.e., pre-defined knowledge), applying an approximate nearest neighbors (ANN) search.

**B. Decision rule module.** By passing through the modules above, the relations between the identifier-linked named entities can be represented as triplets of ($e_0$, $e_1$, *relation*) provided that identifiers are given to genes, proteins, and diseases. We applied several filters for exceptional cases to extract gene-disease relations more closely (See S1 and S2 Texts). After applying these filters, the remaining relations are used for graph visualizations (See Fig 5A). The nodes are connected with edges when there are relations between any two entities (See Fig 5B). If multiple nodes are present as the same identifier, we merge them to form a linked unit (See Fig 5C). In the final layer, the output is passed to the softmax function, and the model predicts two entities have link if the value is greater than 0.5.

**C. Integration module.** Once the model determines the link between the paired entities of Gene and Disease, the output can be further reduced to related gene-disease pairs (See Fig 6A). In detail, we built the database to store the results from multiple documents for reference. The SCREENER web service shows the database with visualization as graphs. For example, when users enter multiple scientific papers as input, the SCREENER web service collects the related gene/protein-disease pairs for each paper. Finally, we construct integrated relations containing two node types (i.e., Gene and Disease) and relations using the information from the multiple input documents (See Fig 6B). In addition to inputting multiple PubMed IDs, the users can also query the text such as 'cancer' and 'COVID19'.

## Training

We use the Pytorch [30], allennlp [31], huggingface [32], and scispacy [33] libraries and annotated abstract data for training. We trained the model using 5-fold cross-validation to determine which gene and disease pairs are related. Furthermore, we set the AdamW [34] optimizer with the learning rate warm up of 0.1 and the initial learning rate of 6e-5.

## Results

We conducted three studies to evaluate the model performance: baseline comparisons of pre-training models, the effect of direct edge-to-model performance, and benchmark comparisons

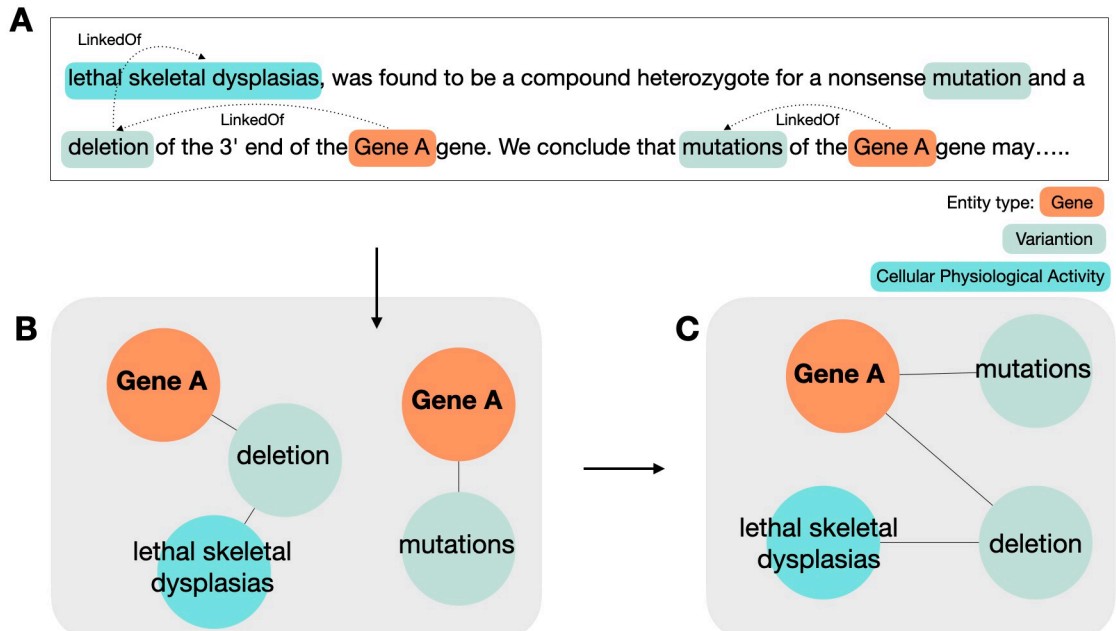

**Fig 5. Merge and reduce graph.** Knowledge graph representation of prediction results NER and RE results from the input text (A) can be represented in the form of a knowledge graph (B). When there are two or more nodes with the same identifier after EL, those nodes are merged into a single node (C).

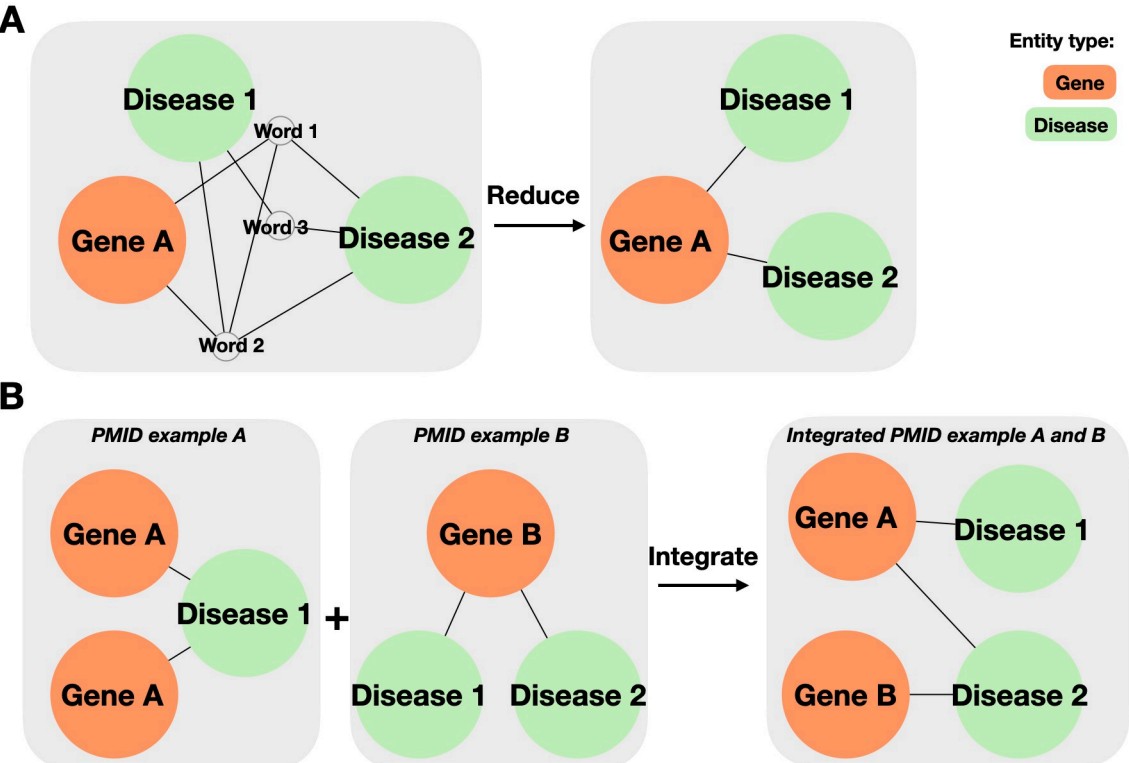

**Fig 6. Details of integration module.** When integrating the results of each pmid, the SCREENER (A) reduces trigger word nodes and (B) integrates gene/protein nodes and disease nodes with others.

**Table 1. SCREENER performance comparison by pre-trained models.**

| Model | Precision | Recall | F1 |
|---|---|---|---|
| PubMedBERT | 0.682±0.04 | 0.739±0.03 | **0.709±0.04** |
| SciBERT | 0.665±0.06 | **0.755±0.01** | 0.706±0.03 |
| BioBERT | 0.709±0.05 | 0.678±0.03 | 0.693±0.03 |
| BioLinkBERT | 0.723±0.04 | 0.627±0.02 | 0.671±0.03 |
| BioELECTRA | **0.730±0.04** | 0.615±0.02 | 0.667±0.03 |
| PubMedBERT_onlyRE | 0.857±0.01 | 0.789±0.02 | 0.821±0.02 |
| SciBERT_onlyRE | 0.846±0.03 | **0.827±0.04** | **0.835±0.02** |
| BioBERT_onlyRE | 0.854±0.02 | 0.748±0.02 | 0.797±0.01 |
| BioLinkBERT_onlyRE | **0.874±0.03** | 0.691±0.02 | 0.771±0.01 |
| BioELECTRA_onlyRE | **0.874±0.03** | 0.729±0.02 | 0.795±0.02 |

on state-of-the-art relation extraction models for predicting disease-target relations. The SCREENER collaboratively trains NER and RE modules, making it difficult to compare with the other relation extraction methods. Consequently, we use the term "onlyRE" to indicate the model that predicts relations only using ground-truth labels of name entities. We added this task to validate the performance of the relation extraction model. We reported mean and standard deviations across 5-fold cross-validation for the performance.

## Pre-trained model

In the past, pre-trained language models (LMs) have been successful in NLP tasks. Recent pre-trained LMs rely on deep neural networks trained with extensive datasets and self-supervised learning, such as masked language modeling [35]. Pre-trained LMs learn various types of knowledge from extensive text training data and produce informative representations to solve NLP problems [36] such as NER and RE. To deal with challenging biomedical NLP problems, many researchers pre-trained LMs using PMC documents. We compared state-of-the-arts pre-training models including PubmedBERT [36], SciBERT [22], BioBERT [7], BiolinkBERT [37], and BioELECTRA [38]. Among these models, SciBERT was one of the best models, hence used for fine-tuning. In detail, we chose SciBERT over PubMedBERT due to its higher recall score and fewer false negatives predictions (See Table 1).

## An effect for adding direct edge (DE)

We conducted the following case study to examine the effect of adding a DE to the model performance. Direct edge represents a edge that connects a gene and a disease directly, in addition to indirect edges of trigger entity (See Fig 4). We evaluated two models, namely SCREENER and SCREENER_onlyRE, in two different training settings (with and without DE). Here, the SCREENER_onlyRE model refers to the model that only performs the relation extraction learning, replacing collaborative learning. For benchmarking purposes, the ground-truth name entities are given to the model, removing the need to train the NER task. For the training environment, we validated the effect of adding a DE in addition to indirect edges. As shown in Table 2, including DE in training improves the model performance. Furthermore, as expected, providing the ground truth labels for name entities enhanced the model performance significantly. The addition of prior knowledge reflects correct identification of name entities plays a significant role in relation extraction model performance.

In detail, the SCREENER and SCREENER_onlyRE models with DE improve the model performance by 4.5 and 4 percentage points (pp) in F1-score, respectively, compared to the

**Table 2. Performance comparison of the SCREENER and SCREENER_onlyRE by using different training setting.**

|  | Precision | Recall | F1 |
|---|---|---|---|
| SCREENER with DE | 0.640±0.03 | 0.909±0.02 | 0.751±0.02 |
| SCREENER without DE | 0.665±0.06 | 0.755±0.01 | 0.706±0.03 |
| SCREENER_onlyRE with DE | 0.803±0.02 | **0.963±0.01** | **0.875±0.01** |
| SCREENER_onlyRE without DE | **0.846±0.03** | 0.827±0.04 | 0.835±0.02 |

models without DE. In both cases, the precision scores are in-par, whereas recall scores are significantly higher in the models with DE. This result shows that adding a DE is highly important in reducing false negatives, i.e., having gene-disease relation, yet the model failed to identify.

## Model benchmark comparisons

For the benchmark, we used the SCREENER_onlyRE model, which predicts gene-disease relations when name entities are given. We compared SCREENER_onlyRE with two outperforming gene-disease relation extraction models: BioBERT-GAD and RENET2 [7, 17]. BioBERT-GAD is a language model trained on large-scale biomedical corpora using BERT architecture. RENET2 is an ensemble of 10 different RE models using a convolutional neural network (CNN) for word representation and a recurrent neural network (RNN) for sentence representation. For model evaluation, we followed the 5-fold cross-validation method presented by RENET2. The detailed process for the benchmark evaluation method is described in S1–S3 Figs, and S2 Table.

Overall, SCREENER_onlyRE outperforms BioBERT-GAD and RENET2 on F1-score by 33.8 and 16.9 percentage points (pp), respectively (See Table 3). The result shows that the SCREENER_onlyRE model can learn underlying gene-disease relations with high specificity compared to the state of the art model. Also, The recall score of SCREENER_onlyRE shows significant improvements of 51.9 and 29.3 pp compared to Biobert and RENET2. One reason for this performance improvement is that the SCREENER applies attention mechanism on predicted gene-disease entity pairs from spans in the NER module, enabling a richer understanding of the context (See Fig 4). Additionally, an information flow from the NER module, i.e., max-pooling and width embedding of the span, and the attention module in the RE module provide a multi-dimensional embedding representation of input texts for relation extraction tasks (See Fig 2). Lastly, adding a direct edge in training enabled the model to learn with higher sensitivity in identifying correct gene-disease relations.

## Discussion

Although previous models successfully identify gene-disease relations in a given document, text mining has critical limitations. BioBERT-GAD extracts the relations of a single type (e.g., disease-gene interactions) at the sentence level, restricting the model from identifying relations

**Table 3. Performance comparison of SCREENER_onlyRE with outperforming RE models: BioBERT-GAD and RENET2.**

|  | Precision | Recall | F1 |
|---|---|---|---|
| BioBERT-GAD | 0.682±0.02 | 0.444±0.01 | 0.537±0.05 |
| RENET2 | 0.748±0.02 | 0.670±0.03 | 0.706±0.01 |
| SCREENER_onlyRE | **0.803±0.02** | **0.963±0.01** | **0.875±0.01** |

beyond a single sentence. Following Biobert-GAD, RENET2 was introduced, which can extract the relations within a single sentence and across multiple sentences [17]. However, RENET2 requires name entities of input genes and diseases in order to predict their relations. Without user input, the model cannot distinguish if a word is a gene or a disease.

Here, we present a pre-trained document-level NLP model that collaboratively learns NER and RE tasks. We further fine-tune the model with gene-disease triplet data using the best-performing pretrained model SciBERT [22]. Using attention mechanisms, the model learns the relative importance of each word in sentences by summing the multiplications of each input with its corresponding attention weight. In addition, the model connects the partner words that commonly appear together using attention weights, discovering gene-disease relations document-level. The model is further enhanced by supplying the additional feature of a direct edge between gene and disease.

The SCREENER is an end-to-end pipeline that learns NER and RE jointly by sharing loss. The model's novelty lies in adding a direct edge in addition to indirect edges in training. Furthermore, attention mechanism applied to predicted gene-disease pairs from spans in the NER module significantly improves the model performance with F1-score of 0.875. Nonetheless, there exist challenges; If the NER model task fails to identify two entities correctly, the RE model inevitably fails to find a relation connecting the two. The most simple yet powerful method to enhance the model is by supplying more data. For example, there have been efforts of document-level annotations for biomedical relations extractions such as BioRED and TBGA [39, 40]. In addition, BigBio data was released as biomedical NLP at Neurips 2022 datasets, and benchmarks track [41], which contains multiple types of entity-to-entity biological relations.

## Conclusion

In the field of fast-paced publications in biomedical literature, an ability to quickly identify gene-disease relations is highly desirable. Yet, manually reading and linking gene-disease relations is arduous as it requires domain knowledge and is time-consuming. To facilitate the literature review, we developed a pre-trained text mining model that rapidly identifies gene-disease relations given keywords of interest or PubMed ids. Furthermore, the web service offers graphical visualizations of protein networks to increase the understanding of the model prediction and output interpretations. This streamlined gene-disease relations identification platform will be highly useful for users interested in analyzing gene-disease relations.

For the future direction, we are interested in training recently published biomedical datasets to build a comprehensive model that increases the predictability power in the NER module, leading to higher accuracy of the RE module. Furthermore, it is possible to train other types of relations beyond training on target-disease relations, such as protein-protein, chemical-target, and chemical-disease interactions. In summary, the applications of SCREENER expand to any relation data to uncover the functional mechanisms of humans.

## Supporting information

**S1 Fig. SCREENER dataset entities distribution.** The distribution of each entity in 1,377 documents of SCREENER dataset.
(TIF)

**S2 Fig. The details of evaluation method.** The evaluation method is following these steps. Generate ground truth graph from json file (See Fig 4A). Generate prediction graph from

results of SCREENER (See Fig 4B). Compare two graphs with confusion matrix (See Fig 3).
(TIF)

**S3 Fig. The details of evaluation method.** The details of evaluation method step 3.
(TIF)

**S1 Text. The details of decision rule module.** We apply several filters for exceptional cases to extract gene-disease associations in a more dedicated manner.
(TIF)

**S2 Text. The details of decision rule module.** The details of decision rule module—B.
(TIF)

**S3 Text. Description of EXCERLA data collection and curation workflow.** A detailed data pre-processing steps described into two objectives.
(TIF)

**S1 Table. The number of each entity in 1,377 documents of SCREENER dataset.**
SCREENER dataset consists of 1,377 annotated document files for extracting gene-disease association. It has 52,709 entities(Gene, Disease, NEGREG, CPA, MPA, REG, VAR, POS-REG, PROTEIN, PATHWAY, INTERACTION, ENZYME) and 43,601 relations(LinkedOf). The Detailed number of each entity is shown in Table 1 and distribution of entities is shown in Fig 1.
(TIF)

**S2 Table. The details of evaluation method.** The detail pairs of ground truth and prediction.
(TIF)

## Acknowledgments

We want to thank EXCELRA for curating high-quality gene-disease relations data.

## Author Contributions

**Data curation:** Hee Jung Koo.

**Software:** Minjun Park, Chan Ung Jeong, Young Sang Baik, Dong Geon Lee, Jeong U. Park.

**Supervision:** Tae Yong Kim.

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
