## [Decision Letter · Decision Letter 0]

13 Apr 2023

PONE-D-23-04283SCREENER: Streamlined Collaborative Learning of NER and RE model for discovering Gene-Disease RelationsPLOS ONE

Dear Dr. Kim,

Thank you for submitting your manuscript to PLOS ONE. After careful consideration, we feel that it has merit but does not fully meet PLOS ONE’s publication criteria as it currently stands. Therefore, we invite you to submit a revised version of the manuscript that addresses the points raised during the review process.

We look forward to receiving your revised manuscript.

Kind regards,

Nguyen Quoc Khanh Le

Academic Editor

PLOS ONE

Journal Requirements:

3. We note that Figure 1 in your submission contain copyrighted images. All PLOS content is published under the Creative Commons Attribution License (CC BY 4.0), which means that the manuscript, images, and Supporting Information files will be freely available online, and any third party is permitted to access, download, copy, distribute, and use these materials in any way, even commercially, with proper attribution. For more information, see our copyright guidelines: http://journals.plos.org/plosone/s/licenses-and-copyright.

Reviewers' comments:

Reviewer's Responses to Questions

**Comments to the Author**

1. Is the manuscript technically sound, and do the data support the conclusions?

Reviewer #1: No

Reviewer #2: Partly

2. Has the statistical analysis been performed appropriately and rigorously? 

Reviewer #1: No

Reviewer #2: I Don't Know

3. Have the authors made all data underlying the findings in their manuscript fully available?

Reviewer #1: Yes

Reviewer #2: No

4. Is the manuscript presented in an intelligible fashion and written in standard English?

Reviewer #1: Yes

Reviewer #2: Yes

5. Review Comments to the Author

Reviewer #1: The authors describe the developed complex tool for discovering Gene-Disease Relations from text,

combining tasks of the named entity recognition, relation extraction and entity linking. The work is not

bad, but the scientific value and novelty do not look significant.

Approaches used for NER and Entity linking are well known. Joint solutions for NER and RE, are already

exists, but authors didn't mention much of them. There is a lack of analysis for the SCREENER corpus. It

is worthwhile to analyze the lexical diversity of it, how does it differ or similar to AGAC in terms of the

disease and genes mentioned in it.The implemented service may be seen only as an addition to the

research article, and not as a result of any study.

The next drawbacks must be corrected.

- The Results and Discussion sections must be centered around the results of the authors' experiments.

They shouldn't contain continuations of the literature review.

-A markup quality of the developed corpus should be evaluated. How many annotators participated, and

what is annotators agreement score?

- Description for figure 2 is required, what are D, G, T. Disease, Gene and trigger?

- Does the NER module extract only the Gene and Disease types, or all 12 types? Same for relations, are

they extracted for 2 entity types or for all entities? If for all, then Indirect Edges connect everyone with

everyone?

- How exactly are entities extracted, was BIO annotation scheme used or span based method?

- "We use a classifier from ELECTRA [19] model for fine-tuning biomedical literature to identify name

entities (NER)". Electra is a language model that encodes text, it does not include a classifier. And what

is meant by "fine-tuning biomedical literature"? How could a literature be fine-tuned ?

- The RE method looks like the most original idea in this paper, but a supporting figure with scheme or a

more detailed description should be provided. First, what method is used to get the entity vector if it

includes multiple tokens? Secondly, it is written that the Max-pooled vector returns "the most likely

relation pair from candidates". How is that? What exactly is fed into the max pooling operation,

embedding vectors of all extracted candidates? Thirdly, it is written that four vectors are concatenated,

including the distance embedding vector between a pair of entities. What pair of entities? Are such

concatenations formed for all possible pairs among candidates? How this concatenated vector is used?

Is it classified by a fully connected layer?

- The authors do not specify how RE loss is calculated and how it is combined with the final loss.

- Was the SCREENER corpus used only in the training, and the test part was from AGAC? Or test subset is

selected from SCREENER corpus?

-Were Direct Edges added to the test set? If yes, then it turns out that when comparing the models with

DE and without DE, there were a different number of relations in the test sets, is it? Or were only the

initial connections considered during evaluation?

-If an open corpus AGAC was used, are there in a literature any results of other researchers on it?

RENET2 and SCREENER tested on onlyRE task for comparison. Again, did they have the same number of

relations to predict, or were DEs added in the test for SCREENER? What task did BioBERT-GAD tested

on? onlyRE or NER+RE?

- If a cross-validation was performed with 5 folds, why no accuracy deviation estimates are given. Are

the given accuracies of 5-folds average? Authors should double-check or explain the different accuracy

scores in the article. The abstract gives f1 0.7789, this result is not in the article. The authors write that

SCREENER_onlyRE outperforms BioBERT-GAD in recall by 32.7%, how was this number obtained if recall

scores are 0.963 and 0.444 respectively? The same with f1 scores.

In the Discussion section said that RENET2 detects correlations in a sentence. What kind of correlations

is in view?

-The conclusion is very weak, since this is a research article, the authors should emphasize what

suggestions they make in their research and what is the scientific novelty of their method.Also authors

declare: "collaborative learning of NER and RE tasks is the model's novelty", but joint models for NER+RE

tasks are already exists: for example:Span-based Joint Entity and Relation Extraction with Transformer

Pre-training - Markus Eberts & Adrian Ulges, End-to-end Neural Coreference Resolution - Kenton Lee et

al., Selivanov A. et al. Relation Extraction from Texts Containing Pharmacologically Significant

-Information on base of Multilingual Language Models, Sboev A. et al. Accuracy Analysis of the End-to-

End Extraction of Related Named Entities from Russian Drug Review Texts by Modern Approaches

Validated on English Biomedical Corpora //Mathematics. – 2023. – Т. 11. – №. 2. – С. 354.).

The novelty may lie either in the distinctive features of the proposed method, which distinguish it from analogues,

but then they need to be highlighted, or in its application to a new problem.

The manuscript is not ready to print and must be improved.

Reviewer #2: In this paper, the authors present a new transformers-based machine learning model to perform NER for gene and disease entities and RE for gene-disease relations from the biomedical literature.

Relation extraction is performed on a document level and a new joint learning objective for NER and RE is proposed. Evaluation is conducted on a newly annotated dataset and compared to two solid baselines. The results seem promising but the manuscript misses several key details to perform a realistic assessment.

Major issues

I have two major issues with the paper: Once with the description of the relation extraction module in the methods section and once with the evaluation setting.

- How does the Relation Extraction Module work exactly?

- BIOBERT also uses an attention module with the [CLS] token for classifying the relation. How does the proposed attention module differ from this? Here, concrete explanations of how the attention module is implemented are missing.

- You mention that you are also using direct edge connections between entities in addition to indirect ones? How are the direct edges incorporated in the relation extraction module exactly? How does it differ from the indirect ones?

- How is the NER loss integrated into the RE loss for end-to-end training? This is not clear from the text.

- Building upon above, how does the relation extraction module help facilitate the cross sentence extraction aspect?

- Some ablation studies would be helpful for this.

- E.g., apply your method only on sentences vs whole documents? Then, compare the results.

- The evaluation settings need more details. It is not really clear on which datasets evaluation is conducted on and whether both the competitors and the proposed methods are trained/fine-tuned on it.

- E.g., in the result tables 2 and 3 it is not clear which dataset is evaluated, the AGAC or the new SCREENER dataset?

- If evaluation is on the SCREENER dataset, are both competitors also trained on it or on their respective datasets (BioBERT on GAD and RENET2 on the RENET2 dataset)?

- Is it actually in-corpus evaluation (so all datasets are trained and evaluated on SCREENER) and not cross-corpus for BioBERT and RENET2 (they are evaluated on a different dataset than they are trained on)?

- If all methods are trained and evaluated in a comparable way (all in-corpus), then the results table need some more discussion.

- There is a large improvement in F1-score reported for the SCREENER model compared to the baselines, where does it come from? You've reported effects from adding direct edges, maybe add some ablation studies for other effects and discuss their influence.

Minor issues

- Some more datasets for evaluation would be nice to have as you already mentioned in the discussion.

- Why do you not also evaluate on the same dataset as RENET2 for comparison? (Or the GAD dataset for BioBERT?) This would allow for more direct comparison of the results.

- GDA is also a widely used benchmark dataset. It would be nice to also evaluate on that.

- Paper: RENET: A Deep Learning Approach for Extracting Gene-Disease Associations from Literature (Wu et al., 2019)

- Have a look at some current state-of-the-art models for the GDA dataset given in: Document-level Relation Extraction as Semantic Segmentation, (Zhang et al., 2021) and SAIS: Supervising and Augmenting Intermediate Steps for

Document-Level Relation Extraction, (Xiao et al., 2021). It would be nice if the results could also be compared to some of those models.

- Regarding your newly annotated dataset, some more information regarding the annotation quality would be helpful.

- How many annotators worked on the dataset? What was the inter-annotator agreement? How were conflicts during annotation resolved?

- What were annotation guidelines of the dataset?

- In section material and methods, maybe rename SCREENER web service to SCREENER entity linking and visualization web service. This makes it more clear belonging to the methods section.

- I do not understand some percentage point calculations in the results, see lines 181 and 201.

- Are they measured in percentage points (pp) or percent? Please check this.

- An additional ablation study for the NER module would be nice to have.

- How well does this perform in isolation, e.g., in comparison to BioBERT?

6. PLOS authors have the option to publish the peer review history of their article (what does this mean?). If published, this will include your full peer review and any attached files.

Reviewer #1: No

Reviewer #2: No

---

## [Author Response · Author response to Decision Letter 0]

8 Jun 2023

Point-by-point rebuttal letter

Paper: SCREENER: Streamlined Collaborative Learning of NER and RE model for discovering Gene-Disease Relations

May 29th, 2023

Reviewer #1

The Results and Discussion sections must be centered around the results of the authors' experiments. They shouldn't contain continuations of the literature review.

We moved the writings on continuations of the literature in the Discussion section to the Conclusion section.

A markup quality of the developed corpus should be evaluated. How many annotators participated, and what is annotators agreement score?

In total, 9 annotators participated. If two annotators agree, then the relation between entities is confirmed. Otherwise, a third annotator gets involved for the final discussion. Detailed information is available at “SCREENER_curation_protocol.pdf” in Zenodo. 

Description for figure 2 is required, what are D, G, T. Disease, Gene and trigger?

D, G, T represent Disease, Gene, and Trigger, respectively. For clarity, we updated the Figure in the revised manuscript.

Does the NER module extract only the Gene and Disease types, or all 12 types? Same for relations, are they extracted for 2 entity types or for all entities? If for all, then Indirect Edges connect everyone with everyone?

The NER module recognizes entities of all 12 types. In contrast, the relation extraction module extracts only 2 entity types (Gene and Disease types).

How exactly are entities extracted, was BIO annotation scheme used or span based method?

Entities are extracted using span-based method. 

"We use a classifier from ELECTRA [19] model for fine-tuning biomedical literature to identify name entities (NER)". Electra is a language model that encodes text, it does not include a classifier. And what is meant by "fine-tuning biomedical literature"? How could a literature be fine-tuned ? 

In detail, we used sub-module of ELECTRA as a classifier, which consists of the following layers: Linear – GELU – dropout – Linear. For clarity, we revised “fine-tuning biomedical literature” to “Fine-tuning using scientific texts from PubMed database.”

The RE method looks like the most original idea in this paper, but a supporting figure with scheme or a more detailed description should be provided. First, what method is used to get the entity vector if it includes multiple tokens? Secondly, it is written that the Max-pooled vector returns "the most likely relation pair from candidates". How is that? What exactly is fed into the max pooling operation, embedding vectors of all extracted candidates? Thirdly, it is written that four vectors are concatenated, including the distance embedding vector between a pair of entities. What pair of entities? Are such concatenations formed for all possible pairs among candidates? How this concatenated vector is used? Is it classified by a fully connected layer?

We drew an additional figure (See Fig. 2) that describes NER and RE modules in detail. To answer the questions in order, 1) we adopted a span-based method to get the entity vector, and 2) max-pooling returns the highest token within the span, extracting the most prominent constituting token to identify an entity. We changed the sentence “the most likely relation pair from candidates” to “the most likely entity that relates to the entity,” which correctly describes the function of max pooling, and 3) pair of entities are predicted entities from the NER module, which consists of max-pooled vector and width embedding. The distance embedding represents the max-pooled vector among tokens between two entities. The concatenated vector is then passed to a neural network classifier that predicts whether two entities are linked. To elaborate on the entire process, we added Figure 2 to visualize model architecture. In the revised manuscript, please refer to line 281 for the RE module explanation.

The authors do not specify how RE loss is calculated and how it is combined with the final loss. 

RE loss is calculated using binary cross entropy. This loss is added to the loss calculated in the NER module. We specified the loss function used in the manuscript.

Was the SCREENER corpus used only in the training, and the test part was from AGAC? Or test subset is selected from SCREENER corpus? 

The final version of the data combines both the SCREENER corpus and AGAC. The edges in the AGAC corpus (“ThemeOf” and “CausedOf”) are changed to “LinkedOf” to match the edge type of the SCREENER corpus.

Were Direct Edges added to the test set? If yes, then it turns out that when comparing the models with DE and without DE, there were a different number of relations in the test sets, is it? Or were only the initial connections considered during evaluation? 

The direct edge is what both models optimize to predict. Hence, the test set stays the same in both cases (with and without DE), as the task for both models is to predict a link between Gene and Disease without prior knowledge. The model with DE optimizes the model with additional edge information in training.

If an open corpus AGAC was used, are there in a literature any results of other researchers on it?

RENET2 and SCREENER tested on onlyRE task for comparison. Again, did they have the same number of relations to predict, or were DEs added in the test for SCREENER? What task did BioBERT-GAD tested on? onlyRE or NER+RE?

As part of BioNLP Open Shared Tasks 2019, there are results on the data, such as the BERT model in “Biomedical relation extraction with pre-trained language representations and minimal task-specific architecture.” As noted previously, the same number of relations are used for prediction. For a fair comparison, BioBERT-GAD is also tested on onlyRE.

If a cross-validation was performed with 5 folds, why no accuracy deviation estimates are given. Are the given accuracies of 5-folds average? Authors should double-check or explain the different accuracy scores in the article. The abstract gives f1 0.7789, this result is not in the article. The authors write that SCREENER_onlyRE outperforms BioBERT-GAD in recall by 32.7%, how was this number obtained if recall scores are 0.963 and 0.444 respectively? The same with f1 scores. In the Discussion section said that RENET2 detects correlations in a sentence. What kind of correlations is in view? 

In the revised manuscript, we reported a mean and standard deviation of 5-fold. We also corrected the F1-score in the abstract, which was the initial performance of SCREENER before optimization. 

We intended to compare the results in terms of percentage points (pp). We modified the sentences (lines 181 and 201) to clarify the performance comparisons between models. 

Furthermore, the term “correlations” is changed to “extracts the relations...” where the word “correlations” was misused and should be corrected.

The conclusion is very weak, since this is a research article, the authors should emphasize what

suggestions they make in their research and what is the scientific novelty of their method. Also authors declare: "collaborative learning of NER and RE tasks is the model's novelty", but joint models for NER+RE tasks are already exists: for example:Span-based Joint Entity and Relation Extraction with Transformer Pre-training - Markus Eberts & Adrian Ulges, End-to-end Neural Coreference Resolution - Kenton Lee et al., Selivanov A. et al. Relation Extraction from Texts Containing Pharmacologically Significant. 

We believe the novelties of SCREENER are 1) attention module added in RE module and 2) addition of direct edges in training. We included the papers mentioned in the comments in our manuscript for the reference models yet emphasizing the novelties mentioned above.

Information on base of Multilingual Language Models, Sboev A. et al. Accuracy Analysis of the End-to-End Extraction of Related Named Entities from Russian Drug Review Texts by Modern Approaches Validated on English Biomedical Corpora //Mathematics. – 2023. – Т. 11. – №. 2. – С. 354.).

The research papers mentioned above are cited as part of the previous efforts to develop end-to-end NER and RE modules.

The novelty may lie either in the distinctive features of the proposed method, which distinguish it from analogues, but then they need to be highlighted, or in its application to a new problem.

The manuscript is not ready to print and must be improved.

We highlighted distinctive features (addition of direct edges) and the effect of the attention module in the manuscript.

Reviewer #2

Major issues

 I have two major issues with the paper: Once with the description of the relation extraction module in the methods section and once with the evaluation setting.

How does the Relation Extraction Module work exactly?

In the revised manuscript, we added Figure 2, which explains the architecture of NER and RE in greater detail. In writing, we provided a detailed explanation of the RE model for clarity. Please refer to line 281 for the detailed explanation of the RE module.

BIOBERT also uses an attention module with the [CLS] token for classifying the relation. How does the proposed attention module differ from this? Here, concrete explanations of how the attention module is implemented are missing.

The significant difference between SCREENER and BIOBERT is the inputs to the classifier. In the case of SCREENER, max-pooled value and width embeddings from the span-based method are added for the classification task. The overall architecture is available in Figure 2 of the revised manuscript. 

You mention that you are also using direct edge connections between entities in addition to indirect ones? How are the direct edges incorporated in the relation extraction module exactly? How does it differ from the indirect ones?

In the training stage, the model is given an additional task to predict the direct edge that connects Gene and Disease entities. We show this additional task in the training process enhances the model performance. (See Table 2 in the manuscript).

How is the NER loss integrated into the RE loss for end-to-end training? This is not clear from the text.

The NER loss is added to the RE loss at the end of each training iteration. We provided Figure 2 in the revised manuscript for clarity (line 229).

Building upon above, how does the relation extraction module help facilitate the cross sentence extraction aspect?

In short, span-based relation extraction treats tokenized words beyond a single sentence, enabling cross-sentence learning. 

Some ablation studies would be helpful for this.

For the ablation study, we examined the effects of adding direct edges in training data.

E.g., apply your method only on sentences vs whole documents? Then, compare the results.

In the paper, we primarily focused on validating the effect of adding a direct edge and attention module in the RE module. The prominence of the span-based method is well-described in the paper “Span-based Joint Entity and Relation Extraction with Transformer Pre-training,” which tokenizes tokens beyond a single sentence.

The evaluation settings need more details. It is not really clear on which datasets evaluation is conducted on and whether both the competitors and the proposed methods are trained/fine-tuned on it.

In summary, the AGAC dataset and SCREENER corpus are combined. We performed 5-fold cross-validations and reported the average and standard deviation in the revised manuscript. We clarified the evaluation settings in the revised manuscript (line 326).

E.g., in the result tables 2 and 3 it is not clear which dataset is evaluated, the AGAC or the new SCREENER dataset?

In tables 2 and 3, a combined data of the AGAC and SCREENER corpus is used.

If evaluation is on the SCREENER dataset, are both competitors also trained on it or on their respective datasets (BioBERT on GAD and RENET2 on the RENET2 dataset)?

Both competitors used the combined dataset in 5-fold cross-validation settings.

Is it actually in-corpus evaluation (so all datasets are trained and evaluated on SCREENER) and not cross-corpus for BioBERT and RENET2 (they are evaluated on a different dataset than they are trained on)?

As mentioned above, both competitors are evaluated in-corpus fashion.

If all methods are trained and evaluated in a comparable way (all in-corpus), then the results table need some more discussion. There is a large improvement in F1-score reported for the SCREENER model compared to the baselines, where does it come from? You've reported effects from adding direct edges, maybe add some ablation studies for other effects and discuss their influence.

We argue the reason for seeing a higher performance with the SCREENER is that the model utilizes trigger words, enabling a richer understanding of the context. Furthermore, an information flow from the NER module (max-pooling + width embedding) and the attention module provides a more delicate embedding representation of the scientific texts for relation extraction tasks. Figure 2 in the revised manuscript will be highly useful for the readers. Lastly, adding a direct edge in training enabled the model to learn with higher sensitivity in identifying correct gene-disease relations.

Minor issues

 - Some more datasets for evaluation would be nice to have as you already mentioned in the discussion.

For the future directions, it will be interesting to see the model performance on different fields of data. To stay within the scope of biomedical language model, we combined AGAC data and SCREENER corpus.

Why do you not also evaluate on the same dataset as RENET2 for comparison? (Or the GAD dataset for BioBERT?) This would allow for more direct comparison of the results.

For the benchmark comparison, we mainly focused on the combined dataset of the AGAC and SCREENER corpus across different models and applied the same validation settings for fair comparisons.

GDA is also a widely used benchmark dataset. It would be nice to also evaluate on that.

Since the SCREENER corpus is also a gene-disease associations corpus from PubMed articles, we did not further use GDA for benchmarking. 

Paper: RENET: A Deep Learning Approach for Extracting Gene-Disease Associations from Literature (Wu et al., 2019)

We have cited RENET2, which was published after RENET.

Have a look at some current state-of-the-art models for the GDA dataset given in: Document-level Relation Extraction as Semantic Segmentation, (Zhang et al., 2021) and SAIS: Supervising and Augmenting Intermediate Steps for Document-Level Relation Extraction, (Xiao et al., 2021). It would be nice if the results could also be compared to some of those models.

It would be interesting to compare our model against the models suggested. For benchmark purposes, we believe showing the performances of RENET2 and BioBERT-GAD is also sufficient.

Regarding your newly annotated dataset, some more information regarding the annotation quality would be helpful.

We supplied the details of annotation methods, e.g., annotation scores and information about annotators. Please see Supplemental File 3 for the details in the revised manuscript.

How many annotators worked on the dataset? What was the inter-annotator agreement? How were conflicts during annotation resolved?

There were 9 annotators for data collection in total. If two annotators agree, then the relation between entities is confirmed. Otherwise, a third annotator gets involved for the final discussion. Detailed information is available at “SCREENER_curation_protocol.pdf” in Zenodo.

What were annotation guidelines of the dataset?

Please see Supplemental File 3 for the details.

In section material and methods, maybe rename SCREENER web service to SCREENER entity linking and visualization web service. This makes it more clear belonging to the methods section.

We appreciate the comments and incorporated the suggested recommendations.

I do not understand some percentage point calculations in the results, see lines 181 and 201. Are they measured in percentage points (pp) or percent? Please check this.

We corrected the term “percent” to “percentage points (pp).” We have rewritten sentences to clarify the performance of SCREENER in comparison to other models.

An additional ablation study for the NER module would be nice to have.

We mainly focused on the effects of 5 different pretrained language models, which was highly useful to select the best performing pretrained model.

How well does this perform in isolation, e.g., in comparison to BioBERT?

Although we have not shown NER_only comparisons, as shown in Table 1, BioBERT shows comparable results to SciBERT (the best-performing pretrained model) in NER+RE and onlyRE tasks.

---

## [Decision Letter · Decision Letter 1]

15 Aug 2023

PONE-D-23-04283R1SCREENER: Streamlined Collaborative Learning of NER and RE model for discovering Gene-Disease RelationsPLOS ONE

Dear Dr. Kim,

Thank you for submitting your manuscript to PLOS ONE. After careful consideration, we feel that it has merit but does not fully meet PLOS ONE’s publication criteria as it currently stands. Therefore, we invite you to submit a revised version of the manuscript that addresses the points raised during the review process.

We look forward to receiving your revised manuscript.

Kind regards,

Nguyen Quoc Khanh Le

Academic Editor

PLOS ONE

Journal Requirements:

Reviewers' comments:

Reviewer's Responses to Questions

**Comments to the Author**

1. If the authors have adequately addressed your comments raised in a previous round of review and you feel that this manuscript is now acceptable for publication, you may indicate that here to bypass the “Comments to the Author” section, enter your conflict of interest statement in the “Confidential to Editor” section, and submit your "Accept" recommendation.

Reviewer #1: All comments have been addressed

Reviewer #3: All comments have been addressed

2. Is the manuscript technically sound, and do the data support the conclusions?

Reviewer #1: Yes

Reviewer #3: Yes

3. Has the statistical analysis been performed appropriately and rigorously? 

Reviewer #1: Yes

Reviewer #3: Yes

4. Have the authors made all data underlying the findings in their manuscript fully available?

Reviewer #1: No

Reviewer #3: Yes

5. Is the manuscript presented in an intelligible fashion and written in standard English?

Reviewer #1: Yes

Reviewer #3: Yes

6. Review Comments to the Author

Reviewer #1: The authors responded to most of the comments and corrected the article accordingly. Still minor revision is needed.

- There is probably a mistake in the phrase "span-based tokens" in the description of Fig 2 . Usually span is the sequence of tokens, how tokens can be span-based.

- "annotation protocol are available at " ext-link-type="uri" xlink:type="simple">https://doi.org/10.5281/zenodo.7445644". Files have closed access, the annotation protocol is not available.

- One of the reasons for the improvemen of f1 score is stated as follows "the SCREENER utilizes trigger word enabling a richer understanding of the context". But it's unclear what does authors mean by that. If SCREENER, RENET2 and BioBERT-GAD were trained and tested on the same edges, they all used edges that connect trigger words with a disease or a gene and that way utilized trigger words. Also other methods build token embeddings considering context around it too.

- "Detailed information on the annotation process is described in Supplemental Text 3". The archive with supplemental materials is not attached to the revision 1.

Reviewer #3: In this paper, the authors present a novel model which learns and extracts document-level relations between genes and diseases. By adding direct and indirect edges between genes and diseases, they improve the performance of the model, and achieve an F1 score of 0.875, which is superior to the other existing approaches. This makes their approach novel and state-of-the-art.

The complementary SCREENER web platform adds to the significance of their approach and the paper overall.

Given that this is a R1 version of the paper, many problematic parts have already been addressed. This leaves the paper at a good state, where no major issues or concerns remain.

The only concern I have is the novelty and importance of this new approach. It is obviously an incremental improvement, but the authors should provide more arguments as to why this is worth publishing in the journal, and not a conference or workshop, for instance.

Other than that, the paper is overall very good, understanable and easy to follow.

7. PLOS authors have the option to publish the peer review history of their article (what does this mean?). If published, this will include your full peer review and any attached files.

Reviewer #1: No

Reviewer #3: No

---

## [Author Response · Author response to Decision Letter 1]

28 Sep 2023

Point-by-point rebuttal letter

Paper: SCREENER: Streamlined Collaborative Learning of NER and RE model for discovering Gene-Disease Relations

September 27th, 2023

Reviewer #1

There is probably a mistake in the phrase "span-based tokens" in the description of Fig 2 . Usually span is the sequence of tokens, how tokens can be span-based.

We agree this was an incorrect description of span. To eliminate confusion, we modified “span- based tokens” to “span.”

"annotation protocol are available at https://doi.org/10.5281/zenodo.7445644". Files have closed access, the annotation protocol is not available.

We changed to open access. The new version is available at: https://zenodo.org/record/8385406.

One of the reasons for the improvement of f1 score is stated as follows "the SCREENER utilizes trigger word enabling a richer understanding of the context". But it's unclear what does authors mean by that. If SCREENER, RENET2 and BioBERT-GAD were trained and tested on the same edges, they all used edges that connect trigger words with a disease or a gene and that way utilized trigger words. Also other methods build token embeddings considering context around it too.

To elaborate on the improvement, the novelty is application of attention mechanism on two entities predicted as Gene and Disease. In the paper, we “SCREENER applies attention mechanism on predicted gene-disease entity pairs from spans in the NER module, enabling a richer understanding of the context.” In comparison, RENET2 does not apply attention mechanism, and BioBERT-GAD does not utilize spans for entity prediction.

"Detailed information on the annotation process is described in Supplemental Text 3". The archive with supplemental materials is not attached to the revision 1.

We attached Supplemental Text 3 in the revision 2. We also attached rest of supplemental files.

Reviewer #2

The only concern I have is the novelty and importance of this new approach. It is obviously an incremental improvement, but the authors should provide more arguments as to why this is worth publishing in the journal, and not a conference or workshop, for instance.

We further elaborated the novelty and importance of the SCREENER in line 394, which states, “The SCREENER is an end-to-end pipeline that learns NER and RE jointly by sharing loss. The model's novelty lies in adding a direct edge in addition to indirect edges in training. Furthermore, attention mechanism applied to predicted gene-disease pairs from spans in the NER module significantly improves the model performance with F1-score of 0.875”. This should reinforce the validity of the model to be published in the journal.

---

## [Editor Report · Decision Letter 2]

8 Nov 2023

SCREENER: Streamlined Collaborative Learning of NER and RE model for discovering Gene-Disease Relations

PONE-D-23-04283R2

Dear Dr. Kim,

We’re pleased to inform you that your manuscript has been judged scientifically suitable for publication and will be formally accepted for publication once it meets all outstanding technical requirements.

Kind regards,

Nguyen Quoc Khanh Le

Academic Editor

PLOS ONE
---

## [Editor Report · Acceptance letter]

13 Nov 2023

PONE-D-23-04283R2 

SCREENER: Streamlined Collaborative Learning of NER and RE model for discovering Gene-Disease Relations 

Dear Dr. Kim:

I'm pleased to inform you that your manuscript has been deemed suitable for publication in PLOS ONE. Congratulations! Your manuscript is now with our production department. 

Kind regards, 

on behalf of

Dr. Nguyen Quoc Khanh Le 

Academic Editor

PLOS ONE